# The Total Denervation of the Ischemic Kidney Induces Differential Responses in Sodium Transporters’ Expression in the Contralateral Kidney in Goldblatt Rats

**DOI:** 10.3390/ijms25136962

**Published:** 2024-06-26

**Authors:** Caroline G. Shimoura, Tales L. Oliveira, Gisele S. Lincevicius, Renato O. Crajoinas, Elizabeth B. Oliveira-Sales, Vanessa A. Varela, Guiomar N. Gomes, Cassia T. Bergamaschi, Ruy R. Campos

**Affiliations:** 1Cardiovascular Division, Department of Physiology, School of Medicine, Federal University of Sao Paulo, Sao Paulo 04023-060, Brazil; carolineshimoura@gmail.com (C.G.S.); gslincevicius@gmail.com (G.S.L.); guiomar.gomes@unifesp.br (G.N.G.); bergamaschi.cassia@unifesp.br (C.T.B.); 2Faculty of Medicine, Municipal University of São Caetano do Sul, Sao Paulo 01327-000, Brazil; tales.oliveira@online.uscs.edu.br; 3Laboratory of Genetics and Molecular Cardiology, Heart Institute, Faculty of Medicine, University of Sao Paulo, Sao Paulo 05508-000, Brazil; renatocrajoinas@gmail.com; 4Faculty of Medicine, Metropolitan University of Santos, Santos 11045-002, Brazil; betholiveira@gmail.com; 5Renal Division, Department of Medicine, School of Medicine, Federal University of Sao Paulo, Sao Paulo 04023-060, Brazil; vanessaavarela@gmail.com

**Keywords:** renovascular hypertension, sympathetic nerve, NHE3 protein, NKCC protein, glomerular filtration rate

## Abstract

The Goldblatt model of hypertension (2K-1C) in rats is characterized by renal sympathetic nerve activity (rSNA). We investigated the effects of unilateral renal denervation of the clipped kidney (DNX) on sodium transporters of the unclipped kidneys and the cardiovascular, autonomic, and renal functions in 2K-1C and control (CTR) rats. The mean arterial pressure (MAP) and rSNA were evaluated in experimental groups. Kidney function and NHE3, NCC, ENaCβ, and ENaCγ protein expressions were assessed. The glomerular filtration rate (GRF) and renal plasma flow were not changed by DNX, but the urinary (CTR: 0.0042 ± 0.001; 2K-1C: 0.014 ± 0.003; DNX: 0.005 ± 0.0013 mL/min/g renal tissue) and filtration fractions (CTR: 0.29 ± 0.02; 2K-1C: 0.51 ± 0.06; DNX: 0.28 ± 0.04 mL/min/g renal tissue) were normalized. The Na^+^/H^+^ exchanger (NHE3) was reduced in 2K-1C, and DNX normalized NHE3 (CTR: 100 ± 6; 2K-1C: 44 ± 14, DNX: 84 ± 13%). Conversely, the Na^+^/Cl^−^ cotransporter (NCC) was increased in 2K-1C and was reduced by DNX (CTR: 94 ± 6; 2K-1C: 144 ± 8; DNX: 60 ± 15%). In conclusion, DNX in Goldblatt rats reduced blood pressure and proteinuria independently of GRF with a distinct regulation of NHE3 and NCC in unclipped kidneys.

## 1. Introduction

The Goldblatt model of hypertension, 2 kidneys–1 clip (2K-1C), in rats has been extensively used to investigate the mechanisms underlying the development and maintenance of elevated blood pressure (BP) in renovascular hypertension [1,2,3]. The reduction in blood flow resulting from the implantation of a silver clip in one of the renal arteries leads to the activation of the renin–angiotensin–aldosterone system (RAAS). Increased circulating RAAS promotes a gradual increase in BP and triggers an increase in renal sympathetic nerve activity (rSNA) that reaches a plateau in the fifth week after hypertension induction [4,5]. Indeed, both the low-frequency power of systolic blood pressure and the RSNA are significantly increased in 2K-1C rats [5]. In addition, either ganglionic blockade or the inhibition of the rostroventrolateral medulla leads to normalization and a larger reduction in blood pressure in 2K-1C rats compared with the control, indicating that sympathoexcitation is an important factor in the maintenance of experimental renovascular-induced hypertension [5]. Furthermore, in human renovascular hypertension, there is a significant increase in muscle sympathetic nerve activity and in the total body spillover of noradrenaline [6]. These observations emphasize the importance and major contribution of sympathetic vasomotor activation in the pathophysiology of renovascular hypertension. Additionally, the synergic interaction between angiotensin II (Ang II) and rSNA contributes to the complexity of renovascular hypertension in 2K-1C rats, inducing various responses, including renal injury, increased oxidative stress and inflammation, reduced renal plasma flow (RPF), an increased glomerular filtration rate (GFR), and changes in sodium balance [7,8,9,10,11].

The rSNA influences renal function by increasing renin secretion, decreasing renal blood flow and the glomerular filtration rate (GFR), and enhancing tubular sodium reabsorption. In fact, inappropriately augmented rSNA plays a crucial role in the onset and maintenance of clinical and experimental arterial hypertension [12,13]. In the consideration of the mechanisms involved in renovascular hypertension, to isolate the effects of renal nerve activation from the BP effects on the kidneys, we previously reported that the acute electrical stimulation of the renal nerve (distal end) in rats increased sodium–hydrogen exchanger 3 (NHE3) expression in the proximal tubule. This effect, induced by renal nerve stimulation, was independent of BP changes and was entirely blocked by prior losartan treatment, which induced a reduction in intrarenal but not angiotensinogen plasma levels. This suggests an interaction between rSNA and local angiotensin II (Ang II) formation in the kidney in sodium balance [14]. The results indicate that, at least acutely, increased rSNA triggers NHE3-mediated sodium reabsorption. However, the way in which this phenomenon operates in the long term with regard to sodium balance remains poorly understood, as do the effects of rSNA on other sodium transporters in the kidney, such as the Na^+^/Cl^−^ cotransporter (NCC) and the epithelial sodium channel (ENaC).

Recently, it has been demonstrated that both total renal denervation and selective afferent renal denervation, as well as the deletion of the transient receptor potential vanilloid 1 channel (TRIPV1) in the kidneys, have attenuated sympathoexcitation, hypertension, and improved renal function in 2K-1C rats [15,16,17,18,19] and in 2K-1C mice. Furthermore, in a previous study, we reported that the total renal denervation of the ischemic kidney leads to the reduction and normalization of sympathetic nerve activity in the contralateral kidney [3,20]. These studies indicate that renal nerves, encompassing both afferent and efferent pathways, play a significant role in initiating kidney dysfunction and autonomic imbalance, consequently contributing to renovascular hypertension.

However, the effects of total renal denervation of the ischemic kidney on renal sodium transporters in the contralateral kidney remain poorly elucidated. Therefore, we postulate that the long-term increase in rSNA contributes to changes in tubular sodium reabsorption throughout the nephron in the contralateral kidney in 2K-1C rats. To test this hypothesis, in the present study, we investigated the effects of the total renal denervation of the clipped kidney on renal function and sodium transporters including NHE3, NCC, and ENaC in the contralateral kidney of 2K-1C renovascular hypertensive rats.

## 2. Results

There was a significant increase in rSNA in both the clipped and unclipped kidneys of the 2K-1C rats compared to the normotensive animals (CTR right kidney: 109.8 ± 7; CTR left kidney: 83.8 ± 6 spikes/s; 2K-1C clipped kidney: 166.2 ± 8; 2K-1C unclipped kidney: 171.4 ± 10 spikes/s). The renal denervation of the ischemic kidney (DNX) significantly decreased rSNA in the contralateral kidney (2K-1C DNX: 95 ± 9, spikes/s vs. 2K-1C: 171.4 ± 10 spikes/s) in 2K-1C DNX group compared to 2K-1C, as shown in Figure 1.

The mean arterial pressure (MAP) (CTR: 101 ± 2; 2K-1C: 175 ± 5 mmHg) and heart rate (HR) (CTR: 316 ± 9; 2K-1C: 392 ± 11 bpm) were significantly increased in the 2K-1C rats compared to the control group (CTR). A significant reduction in MAP (DNX: 146 ± 5 mmHg) but not in HR was induced by DNX (Figure 2G,H). 

### 2.1. Proteinuria Decreases after DNX in 2K-1C Rats

Water intake and urinary flow were significantly increased in the 2K-1C group compared to the CTR group; however, no significant changes were observed in these parameters after the renal denervation of the clipped kidney (Figure 2C,D). The body weights, Na^+^ excretion, and Na^+^ plasma levels were not significantly different among the groups (Figure 2A,E,F). A significant increase in proteinuria was found in the 2K-1C group compared to the CTR group (2K-1C: 102 ± 26 vs. CTR: 21 ± 1 mg/dL). The renal denervation of the clipped kidney significantly reduced proteinuria in the 2K-1C rats (2K-1C DNX: 36 ± 9 mg/dL vs. 2K-1C), as shown in Figure 2B.

### 2.2. GFR and RPF Were Not Affected in Unclipped Kidney by DNX

The GFR was significantly decreased in the 2K-1C group compared to the CTR group (CTR: 0.56 ± 0.09; 2K-1C: 0.26 ± 0.02; 2K-1C DNX: 0.20 ± 0.07 mL/min, Figure 3B), while the RPF did not show significant differences among the groups (CTR: 1.4 ± 0.3; 2K-1C: 0.62 ± 0.13; 2K-1C DNX: 0.81 ± 0.26 mL/min, Figure 3A). DNX did not alter the GFR or RPF in the 2K-1C group. However, DNX caused significant decreases in urinary flow (CTR: 0.0042 ± 0.001; 2K-1C: 0.014 ± 0.003; 2K-1C DNX: 0.005 ± 0.0013 mL/min/g renal tissue) and the filtration fraction rate (CTR: 0.29 ± 0.02; 2K-1C: 0.51 ± 0.06; 2K-1C DNX: 0.28 ± 0.04 mL/min/g renal tissue), as shown in Figure 3C,D.

### 2.3. Effects of Unilateral Renal Denervation on NHE3, NCC, ENaCβ, and ENaCγ Protein Expressions in Unclipped Kidney

DNX of the clipped kidneys significantly increased NHE3 protein expression in the unclipped kidneys of the 2K-1C rats (2K-1C DNX: 84 ± 13 vs. 2K-1C: 44 ± 14%), as shown in Figure 4A. Conversely, NCC protein expression, which was significantly increased in the unclipped kidneys of the 2K-1C rats, was significantly decreased by renal denervation (CTR: 94 ± 6; 2K-1C: 144 ± 8; 2K-1C DNX: 60 ± 15%, Figure 4B). No significant differences were observed in ENaCβ protein expression in the unclipped kidneys (CTR: 100 ± 3; 2K-1C: 81 ± 9; 2K-1C DNX: 77 ± 11%, Figure 4C) or ENaCγ protein expression (CTR: 100 ± 3; 2K-1C: 105 ± 6; 2K-1C DNX: 103 ± 3%, Figure 4D) among the groups. However, total cortical NHE3 protein expression was decreased in the unclipped kidneys of the 2K-1C group compared to the CTR group (CTR: 100 ± 6; 2K-1C: 44 ± 14%, Figure 4A).

## 3. Discussion

Our main findings regarding the effects of unilateral renal denervation of the clipped kidneys in the 2K-1C rats were as follows: (1) a decrease in NCC and normalized NHE3 protein expression in the unclipped kidneys; (2) decreases in the MAP and rSNA; (3) decreases in urinary flow and filtration fraction, and (4) normalized total proteinuria.

The present study shows that, despite the differences in perfusion pressure between the kidneys (ischemic and contralateral kidneys) in the 2K-1C model, sympathetic nerve activity did not show significant variation between the right and left kidneys. This observation aligns with the hypothesis put forth by McAllen and Dampney, which suggests that vasomotor sympathetic activity is organized by the brain in a topographically structured manner. According to this hypothesis, both the right and left kidneys receive sympathetic activity of a similar magnitude, as this activity is organized by specialized neurons preferentially related to the control of sympathetic activity to the kidneys, located in the ventrolateral portion of the medulla oblongata [21].

Interestingly, sympathetic hyperactivity is not limited to the kidneys alone. There is an increase in SNA for the vasculature, heart, and cervical nerve in the Goldblatt model of hypertension [22,23]. Furthermore, the stimulation of renal sensory nerves increases sympathetic efferent activity to splanchnic, lumbar, and renal nerves, inducing a sympathetically mediated increase in the BP of 2K-1C mice [24]. Thus, in the present study, we describe that the total renal denervation of the ischemic kidney decreases sympathetic drive in the contralateral kidney, indicating that afferents from the ischemic kidney play a role in driving sympathetic nerve activity to the contralateral side. This observation is consistent with our previous findings demonstrating that selective afferent renal nerve ablation through topical capsaicin application in the ischemic kidney led to a reduction in rSNA in the contralateral kidney in 2K-1C rats [17]. Furthermore, it has been described that the electrical stimulation of mouse renal afferent nerves resulted in a frequency-dependent increase in BP, which was abolished by ganglionic blockade, suggesting that the activation of renal afferents increases sympathetic nerve activity [24]. Although we have not performed the selective denervation of afferent fibers, the current study shows that the removal of both the afferent and efferent fibers of the ischemic kidney decreases the contralateral rSNA, indicating that in this phase of 2K1C hypertension, afferents from the ischemic kidney play a major role in driving rSNA. The relative contributions of afferent and efferent fibers on sodium transporters remain to be elucidated.

As previously shown and confirmed in the current study, DNX normalized rSNA in the contralateral kidney and reduced but did not normalize BP, suggesting that, in addition to renal sympathoexcitation, other mechanisms contribute to the maintenance of hypertension in 2K-1C rats six weeks after clipping [25]. Indeed, other mechanisms, such as Ang II actions, oxidative stress, inflammation, kidney fibrosis, and the dysfunction of the arterial baroreceptor and chemoreceptor reflexes, play roles in sustaining an elevated BP in renovascular hypertension [26].

The implantation of a silver clip around the renal artery initially leads to a significant reduction in renal RBF, consequently triggering the activation of the RAAS. However, as hypertension progresses, the plasma concentration of Ang II declines, and six weeks after hypertension induction (phase 2), the plasma Ang II concentration becomes almost the same as that in normotensive animals [8]. Nonetheless, it is important to note that during this phase of renovascular hypertension, a significant increase in AT1 receptor expression was found in the RVLM and PVN. The renal denervation of the ischemic kidneys in 2K-1C rats results in a reduction in at1 receptor expression in the brain, which is associated with a significant reduction in oxidative stress in sympathetic premotor neurons (RVLM and PVN), rSNA, and hypertension [3]. Additionally, besides the interaction between the sympathetic system and Ang II in the kidneys, regulating renal function and sodium excretion, it is likely that afferent signals originating from the ischemic kidney induce modifications in the central actions mediated by Ang II in the brain. This hypothesis is supported by the finding that afferent renal nerves are necessary for chronic increases in both water intake and vasopressin release following renal artery stenosis [19]. Thus, afferents from the ischemic kidney lead to brain changes that can trigger distinct mechanisms involved in the generation and maintenance of arterial hypertension, such as increases in salt appetite and water intake, and changes in renal sodium handling.

The decreases in the GFR and PRF in the unclipped kidney, as described in this study, might be attributed to the damage in the microvasculature and renal architecture caused by inflammation and angiogenesis, which are well-characterized phenomena in this model [27]. Indeed, our data show a significant increase in total proteinuria in 2K-1C rats, whereas the denervation of the clipped kidney significantly reduces total proteinuria to normal levels, indicating an amelioration in renal injury. Despite the decrease in the GFR, the filtration fraction in the unclipped kidneys was increased in the 2K-1C rats. It is recognized that an elevation in perfusion pressure can augment the intercellular spaces in the proximal tubule, thereby increasing water and solute transport via paracellular pathways [28,29,30]. A previous study using the 2K-1C model has demonstrated that the GFR in the contralateral kidney hypertrophies increases due to an increase in the single nephron glomerular filtration rate (SNGFR) [31]. We did not evaluate the SNGFR in the unclipped kidney in our study. However, it is important to note that we measured the GRF for the left and right kidneys by evaluating the total kidney function for each kidney. Therefore, we suggest that the SNGRF may increase as a compensatory mechanism, but the total GRF normalized by kidney tissue mass is reduced.

We found a differential pattern for sodium transporter expression along the nephron in the unclipped kidneys of the 2K-1C animals. Considering that NHE3 and NCC are preferentially expressed in the proximal and distal tubules, respectively, we suggest that the renal denervation of the clipped kidney induced preferential changes in sodium transporters along the tubule in the contralateral kidney. In contrast, previous studies have shown that acute renal sympathoexcitation leads to an increase in NHE3 protein expression, which is contrary to the findings in our present study [14,32]. Our hypothesis states that the downregulation of NHE3 in the proximal tubule results in reduced sodium reabsorption, leading to a higher sodium concentration in the tubular fluid entering the distal tubule. Thus, we suggest that the increase in NCC in the distal tubule is a compensatory mechanism due to the decrease in sodium reabsorption in the proximal tubule as hypertension progresses. The result is an adequate balance of sodium and water, which was confirmed by the measurements of plasma and urine sodium concentration that do not show differences between the 2K-1C and control animals. Moreover, it is known that acute increases in BP can trigger the translocation of NCC from the cell membrane to vesicles, reducing sodium reabsorption in the distal tubule [33]. However, in the presence of Ang II, the mechanism differs as NCC remains on the membrane despite the increased BP [34].

Interestingly, our findings indicate that NCC protein expression decreased and NHE3 was normalized after renal denervation in the 2K-1C model. These alterations may be attributed to the normalization of rSNA and the reduction in BP induced by renal denervation. Notably, these changes in sodium transporter expression did not disrupt sodium and water balance, as confirmed by our measurements, indicating that the phase of renovascular hypertension investigated in this study is not volume-dependent, which is consistent with previous research [35].

### The Limitations of the Study

BP was recorded directly and acutely, which may have caused some stress; however, all animals were subjected to the same experimental protocol and under conditions that minimized stress related to instrumentation. Finally, sympathetic activity was directly recorded and compared between groups. Despite being a challenging method with limitations, all recordings were made with the same amplification and filters and were tested for their barosensitivity and response to the ganglionic blocker hexamethonium. In addition, only spikes sensitive to hexamethonium were quantified. Therefore, we consider that the data obtained in the present study are robust enough to support the view that rSNA is elevated in the 2K-1C Goldblatt model of hypertension in rats, confirming the results of previous study [3].

In summary, the results of this study show that the renal nerves of the ischemic kidney play an important role in controlling the function of the contralateral kidney submitted to an elevated BP in 2K-1C rats. The renal denervation of the ischemic kidneys in 2K-1C rats reduced BP and proteinuria independently of the GFR. Additionally, renal nerves play an important role in sodium and water excretion, differentially modulating NHE3 and NCC protein expressions in the unclipped kidney.

## 4. Material and Methods

### 4.1. Animals

All animal procedures complied with the standards for the care and use of experimental animals and were approved by the Animal Ethics Committee (11383710/13) of the Federal University of Sao Paulo (UNIFESP), Sao Paulo, Brazil. This study is reported in accordance with the Animal Research: Reporting of In Vivo Experiments (ARRIVE) guidelines [36]. Male Wistar rats (150–180 g) were purchased from Central Animal House (CEDEME) at UNIFESP. The animals were group-housed, provided with unrestricted access to rat chow and water, and maintained under a 12 h light/dark cycle in a temperature-controlled environment. Male Wistar rats (*Rattus norvegicus*) weighing between 150 and 180 g were used for the induction of renovascular hypertension, and those weighing between 280 and 300 g were used as controls. All animals were purchased from the Central Animal Facility (CEDEME) at UNIFESP. The animals were maintained under the following conditions: (a) a 12 h light/dark cycle; (b) an ambient temperature of 23 ± 2 °C; and (c) ad libitum access to water and rodent chow. The rats were divided into three groups according to the experimental protocols: (1) CTR: control rats; (2) 2R1C: hypertensive rats; and (3) 2R1C DNX: hypertensive rats subjected to unilateral denervation (clipped kidney). The hypertensive rats were randomly separated into the 2K1C and 2K1C DNX groups before the DNX surgery.

In this study, we conducted three independent experimental series involving both normotensive models and 2K-1C Goldblatt models in rats: (1) The first series aimed to concurrently assess cardiovascular and autonomic functions, as well as renal sympathetic nerve activity (rSNA), in both clipped and unclipped kidneys. (2) The second series aimed to evaluate renal parameters, including the glomerular filtration rate (GFR) and renal plasma flow (RPF), by utilizing para-aminohippuric acid and the inulin clearance method, specifically in the unclipped kidneys. (3) The third series was designed to investigate renal transporters (NHE3, NCC, and ENaC) in the unclipped kidneys through a Western blot analysis.

Briefly, the rats were intraperitoneally (i.p.) anesthetized using ketamine (90 mg/kg; Syntec, Cotia, Brazil) and xylazine (10 mg/kg; Syntec, Cotia, Brazil). The left kidney was exposed via a ventral incision, and the left renal artery was carefully isolated from the surrounding connective tissue and partially occluded by the placement of a silver clip (0.2 mm width). During the four weeks after clip placement in the left renal artery, blood pressure was estimated by tail-cuff plethysmography every week (BP 2000, Visitech Systems, Raleigh, NC, USA). Only rats with tail pressure at ≥180 mmHg were selected to undergo total denervation of the clipped kidney or used as controls.

Total renal denervation was induced 4 weeks after clip placement. The animals were once again i.p. anesthetized with ketamine and xylazine (90 mg/kg and 10 mg/kg, respectively; Syntec, Brazil). A retroperitoneal incision was made to expose the renal pedicle. All nerves surrounding the left renal artery were carefully sectioned, and the left renal artery was painted with 10% phenol in the DNX group. To manage postoperative pain, an intramuscular injection of meloxicam (2.5 mg/kg/day) (Boehringer Ingelheim, Ingelheim, DE, USA) was administered consecutively for three days, and the rats were given a two-week recovery period.

Two weeks after renal denervation, the animals were placed into individual metabolic cages (Nalgene Nunc, Rochester, NY, USA) for 48 h to facilitate urine collection as well as measurements of water intake and urine output. During the initial 24 h, the animals were allowed to acclimate to their new housing conditions, with data collection taking place during the subsequent 24 h. Sodium levels in plasma and urine were measured using a Flame Photometer (Corning 410C, Sherwood Scientific, Cambridge, UK). Protein levels in urine were determined employing the Lowry method with a DC protein assay kit (Bio-Rad Laboratories Ltd., Hemel Hempstead, UK) [37]. Confirmation of renal denervation in the ischemic kidney was confirmed by a reduction in the renal norepinephrine content, as previously reported [5].

### 4.2. Glomerular Filtration Rate (GFR) and Renal Plasma Flow (RPF)

After housing in a metabolic cage, the rats were i.p. anesthetized with thiopental sodium (60 mg/kg, Cristália, Asuncion, Brazil). The body temperature of each animal was maintained at 37 °C using a servo-controlled heating blanket (LE 13206, LSI Letica Scientific Instruments, Woonsocket, RI, USA), and tracheostomy was performed to reduce airway resistance. A polyethylene catheter (PE-50 tubing) was implanted in the femoral artery for recording BP. A PE-50 catheter was surgically implanted into the right carotid artery for blood sample collection. Urine was collected through a 24G needle (3/4” 0.7 × 19 mm) syringe implanted in the right unclipped kidney. A solution containing 5 mg/min/kg of inulin (Sigma Aldrich, St Louis, MO, USA) and 1.33 mg/min/kg of *para*-aminohippuric acid (PAH) (Sigma Aldrich, St Louis, MO, USA) was continuously infused via the femoral vein using a pneumatic pump at a rate of 1.2 mL/h (KDS 100, KD Scientific, Holliston, MA, USA). Three samples of urine and blood were collected at 30 min intervals. GFR was determined through inulin clearance, and RPF was determined by PAH clearance. The values obtained from inulin and PAH represent the means of the three collection periods. The GFR and RPF were normalized per gram (g) of kidney tissue.

### 4.3. Blood Pressure (BP) and Renal Sympathetic Nerve Activity (rSNA) Measurements

In an independent series of experiments, the rats were i.p. anesthetized with ketamine (90 mg/kg) and xylazine (10 mg/kg) (Syntec, Brazil). A polyethylene catheter (PE-50 tubing) was implanted in the femoral vein for drug infusion, and another was placed in the femoral artery for recording BP. After 24 h of recovery, BP and heart rate (HR) were recorded in conscious freely moving rats (PowerLab, AD Instruments, Bella Vista, Australia). Subsequently, the animals were intravenously (i.v.) anesthetized with 1.4 g/kg of urethane (Sigma-Aldrich, St. Louis, MO, USA) through the femoral vein, and tracheostomy was performed to reduce airway resistance. The renal sympathetic nerve was retroperitoneally exposed, dissected, placed in a silver bipolar electrode, and immersed in mineral oil during the experiment. rSNA was amplified (20 K) and filtered at 20 kHz, with a bandwidth of 100–1000 Hz, using Neurolog equipment (Digitimer, Welwyn Garden, UK), and the data were collected using a PowerLab data acquisition system (AD Instruments, Bella Vista, Australia). At the end of the experiments, the background noise level was determined using i.v. hexamethonium bromide (30 mg/kg, Sigma-Aldrich, St. Louis, MO, USA). The nerve activity was analyzed using Spike Histogram software version 2.6.3 (AD Instruments, Bella Vista, Australia). The spikes were quantified above the background noise determined after i.v. hexamethonium bromide administration and were expressed as spikes per second (spikes/second), as previously reported [20].

### 4.4. Western Blotting

Kidney proteins from the unclipped kidney were isolated in RIPA buffer with a protease inhibitor cocktail (Sigma-Aldrich, St. Louis, MO, USA). Total protein quantification was performed using the Lowry method [37]. Equal protein amounts were loaded and separated by electrophoresis on glycine SDS-PAGE gels. Proteins were transferred to 0.45 µm PVDF membranes (Millipore, Burlington, MA, USA) and blocked for 1 h with 5% non-fat dry milk in 1X PBS-T buffer. After blockage, the membranes were incubated overnight with the following primary antibodies: (1) anti-NHE3 (a generous gift from Dr. Peter Aronson (Yale University, New Haven, CT, USA), (2) anti-NCC (a generous gift from Dr. Alicia McDonough), (3) anti-ENaC γ (Alamones Labs, Jerusalem, IL), (4) anti-ENaC β (Alamones Labs, Jerusalem, IL), (5) anti-GAPDH (Abcam, Cambridge, MA, USA), or (6) anti-actin (Merck, Darmstadt, DE). They were also incubated for 1 h with secondary HRP-conjugated antibodies (Abcam). The signal was measured using chemiluminescence with Immobilon Western reagents (Millipore, Burlington, MA, USA). Band densitometry was performed using ImageJ software version 1.53 (NIH, Bethesda, MD, USA).

### 4.5. Statistical Analysis

Statistical analyses were performed using GraphPad Prism software version 5.0 (GraphPad, San Diego, CA, USA). The data are presented as the mean ± standard error of the mean (mean ± SEM). One-way ANOVA followed by Newman–Keuls post hoc test were employed to identify statistical differences between groups. Values with *p* < 0.05 were considered statistically significant.

## 5. Conclusions

The renal denervation of the ischemic kidney induced the normalization of the spike frequency of rSNA in the contralateral kidney and a reduction in BP. These responses were associated with a decrease in NCC protein levels and an increase in NHE3 protein levels independent of the GFR and RPF in the contralateral unclipped kidneys in 2K-1C rats. Thus, our findings suggest that the renal nerve differentially regulates sodium transporters in the contralateral kidney.

### Perspectives

Renovascular hypertensive rats demonstrate the ability to maintain water and sodium homeostasis through adaptations in renal function and sodium reabsorption after six weeks of hypertension. Compensatory mechanisms in renal function and sodium reabsorption help regulate sodium balance despite sympathetic overactivity and an elevated BP. Our study shows that renal nerves have a pivotal role in the development and maintenance of a 2K-1C model of hypertension in rats, with a differential control on sodium transporters, including the decrease in NCC and normalization of NHE3 protein expression in the contralateral kidney. However, further studies are needed to elucidate the long-term effects of rSNA on the intrarenal mechanisms related to sodium and water reabsorption and, consequently, the development of renovascular hypertension.

## Figures and Tables

**Figure 1 ijms-25-06962-f001:**
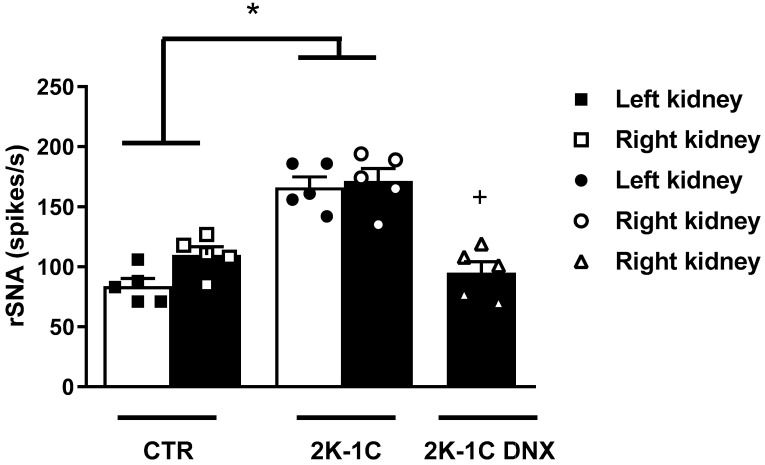
Resting rSNA in left and right kidneys in control and 2K-1C rats. Renal sympathetic nerve activity (rSNA) in spikes per second (spikes/s) in right and left kidneys of control (CTR) and renovascular hypertensive rats (2K-1C), as well as rSNA in right kidneys of 2K-1C rats submitted to renal denervation of clipped kidney. Number of rats used in each group is indicated in bars. Circles, squares and triangles represent individual rats in each group. Values are presented as means ± SEM. * *p* < 0.05 vs. CTR group; ^+^ *p* < 0.05 vs. 2K-1C group (one-way ANOVA followed by Newman–Keuls post hoc test).

**Figure 2 ijms-25-06962-f002:**
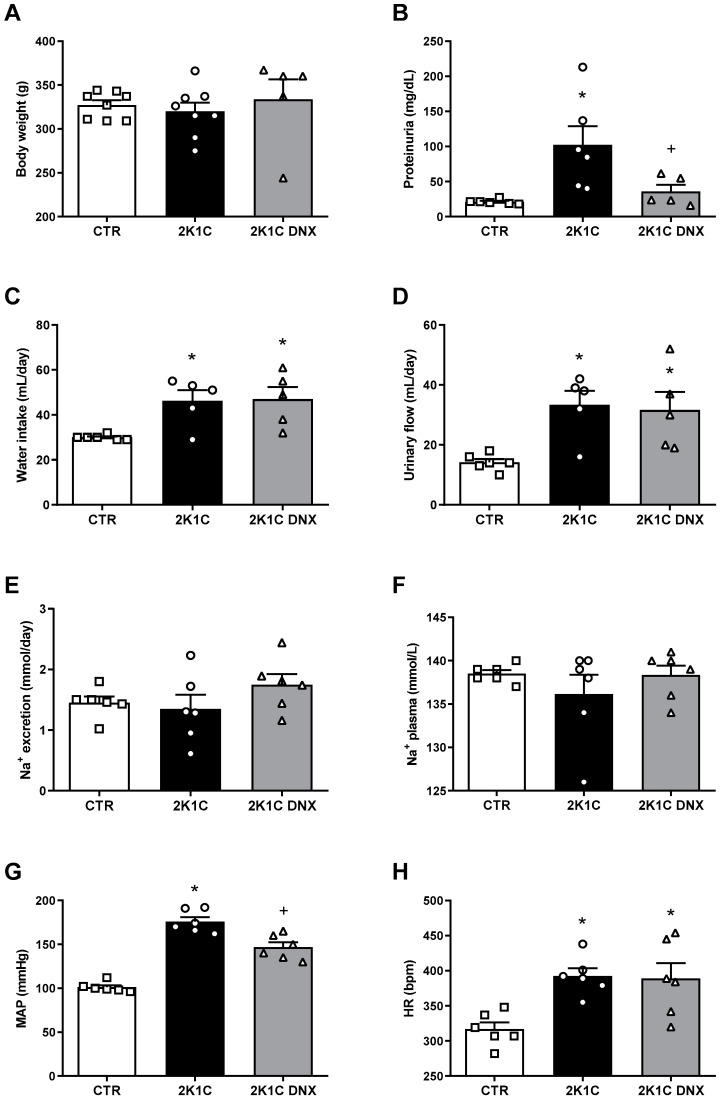
Body weight, water intake, urinary flow, food intake, electrolyte balance, and cardiovascular parameters in control rats, 2K-1C rats, and 2K-1C rats submitted to renal denervation of ischemic kidney. Effects of renal denervation (DNX) on (**A**) Body weight (g): Measurement of body mass in grams. (**B**) Total proteinuria (mg/dL): Level of protein excreted in the urine, indicating kidney function. (**C**) Water intake (mL/day): Daily water consumption in milliliters per day. (**D**) Urinary flow (mL/day): Volume of urine excreted per day. (**E**) Daily Na^+^ excretion (mmol/L/day): Amount of sodium excreted in urine daily. (**F**) Na^+^ plasma (mmol/L): Sodium concentration in plasma. (**G**) MAP (mmHg): Mean arterial pressure in millimeters of mercury. (**H**) HR (bpm): Heart rate in beats per minute. Circles, squares and triangles represent individual rats in each group. Values are presented as means ± SEM. * *p* < 0.05 vs. CTR group; ^+^ *p* < 0.05 vs. 2K-1C group (one-way ANOVA followed by Newman–Keuls post hoc test). Abbreviations: ANOVA, analysis of variance; CTR, control; 2K-1C, 2-kidney 1-clip model; DNX, renal denervation; MAP, mean arterial pressure; HR, heart rate.

**Figure 3 ijms-25-06962-f003:**
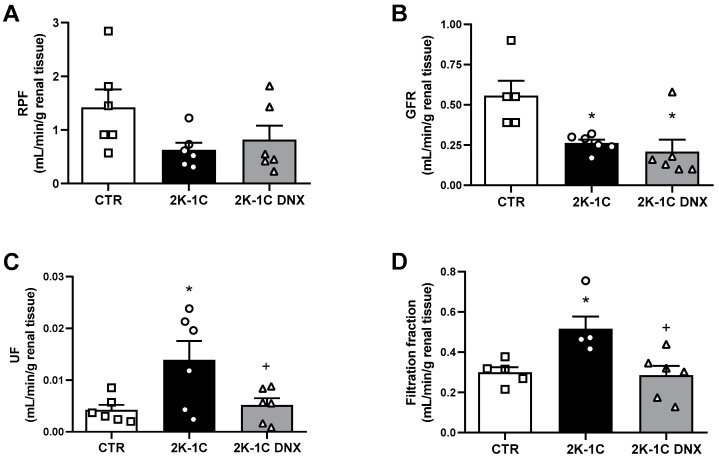
Effects of unilateral renal denervation of ischemic kidney on renal function. (**A**) Renal plasma flow (RPF) (mL/min/g renal tissue), (**B**) glomerular filtration rate (GFR) (mL/min/g renal tissue), (**C**) urinary flow (UF) (mL/min/g renal tissue), and (**D**) filtration fraction (mL/min/g renal tissue). Number of rats used in each group is indicated in bars. Circles, squares and triangles represent individual rats in each group. Values are presented as means ± SEM. * *p* < 0.05 vs. CTR group; ^+^ *p* < 0.05 vs. 2K-1C group (one-way ANOVA followed by Newman–Keuls post hoc test).

**Figure 4 ijms-25-06962-f004:**
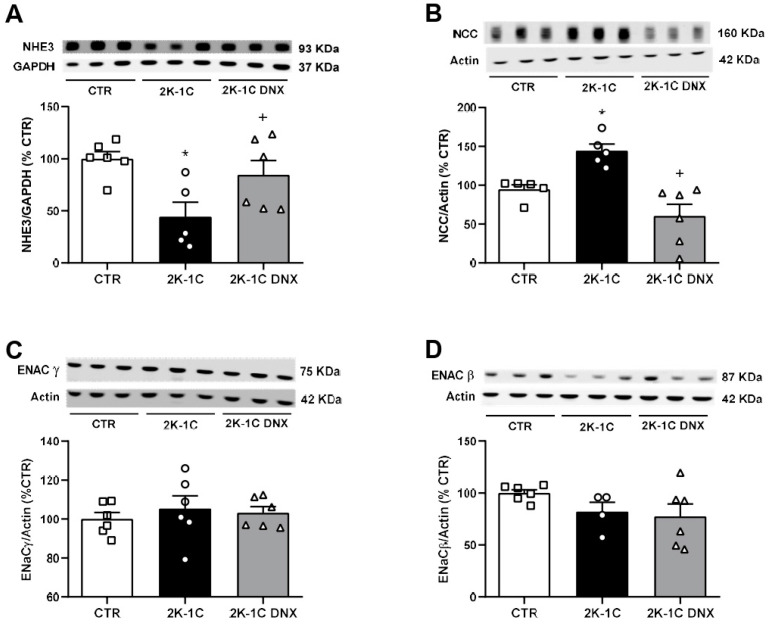
Effects of unilateral renal denervation of ischemic kidney on sodium transporters in unclipped kidney. (**A**) NHE3 protein expression, (**B**) NCC protein expression, (**C**) ENaC γ protein expression, and (**D**) ENaCβ protein expression. Number of rats used in each group is indicated in bars. Circles, squares and triangles represent individual rats in each group. Values are presented as means ± SEM. * *p* < 0.05 vs. CTR group; ^+^ *p* < 0.05 vs. 2K-1C group (one-way ANOVA followed by Newman–Keuls post hoc test).

## Data Availability

The raw data supporting the conclusions of this manuscript will be made available by the corresponding author upon reasonable request.

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
