# Peer review of "The Total Denervation of the Ischemic Kidney Induces Differential Responses in Sodium Transporters’ Expression in the Contralateral Kidney in Goldblatt Rats"

_ijms, 2024, doi:10.3390/ijms25136962_

Round 1

Reviewer 1 Report

Comments and Suggestions for Authors

This is a very interesting study of neurocardiovascular regulation in renovascular hypertension -2K-1C hypertension model. The authors are commended for completing a very arduous task of measuring RSNA on both the left and the right kidney of the rat, that is a tremendous  amount of very technical work. Please find below a few points that I believe would strengthen this manuscript:

1. The hemodynamic aspect of renal denervation in 2K1C has been published in 2023, and in fact you cite that paper, this is ref# 20. I understand that you also had to report these results in order to demonstrate that in your hands, RDX also works. Could you please just comment in the discussion or limitation sections regarding the lack of novelty of your results in this respect? The rest of the data you show is indeed novel, and you should certainly underline that throughout your discussion.

2. One of the conclusions is that despite the difference in perfusion pressure between the clipped and the unclipped kidneys, RSNA was the same on both sides. However, you only show one characteristic of RSNA, which is the spike frequency (in spikes/second). However, there are no data showing the tonic, peak and phasic activity of the bursts, as well as any other burst characteristics (i.e. width 50, Tau etc.). Unless these characteristics of RSNA bursting activity is included in the results section, perhaps re-phrasing that conclusion is in order to state that there is no difference in spike frequency. Lab chart is capable of completing these analyses, and given that there is time and resources, perhaps even adding this data to the manuscript would solidify these comparisons between the clipped and the non-clipped kidney.

3. Could you please include the representative raw tracings of the renal nerve activity (unless there are space constraints). It paints more complete picture when quantified RSNA is accompanied by raw tracings.

4. You completed RDNX 4 weeks after clipping, and not prior to clipping. Could you please include the rationale for this? I understand that there are different aspects of hypertension development and maintenance that one could study. It didn't come through in the manuscript why you elected to follow this timeline and what you were hoping to achieve by doing this as opposed to for example denervating prior to clipping.

Author Response

Reviewer #1

Comments: The This is a very interesting study of neurocardiovascular regulation in renovascular hypertension -2K-1C hypertension model. The authors are commended for completing a very arduous task of measuring RSNA on both the left and the right kidney of the rat, that is a tremendous amount of very technical work. Please find below a few points that I believe would strengthen this manuscript:

Specific comments are as follows:

  1. The hemodynamic aspect of renal denervation in 2K1C has been published in 2023, and in fact you cite that paper, this is ref# 20. I understand that you also had to report these results in order to demonstrate that in your hands, RDX also works. Could you please just comment in the discussion or limitation sections regarding the lack of novelty of your results in this respect? The rest of the data you show is indeed novel, and you should certainly underline that throughout your discussion.

Response: We appreciate your contribution to the clarity of our article. We acknowledge the previous publication of the hemodynamic aspect of renal denervation in the 2K1C model. We included these arguments to validate that the 2K1C model has been successfully reproduced with high quality in our laboratory for a long time. Additionally, we have emphasized the novel findings of our study throughout the discussion.

  1. One of the conclusions is that despite the difference in perfusion pressure between the clipped and the unclipped kidneys, RSNA was the same on both sides. However, you only show one characteristic of RSNA, which is the spike frequency (in spikes/second). However, there are no data showing the tonic, peak and phasic activity of the bursts, as well as any other burst characteristics (i.e. width 50, Tau etc.). Unless these characteristics of RSNA bursting activity is included in the results section, perhaps re-phrasing that conclusion is in order to state that there is no difference in spike frequency. Lab chart is capable of completing these analyses, and given that there is time and resources, perhaps even adding this data to the manuscript would solidify these comparisons between the clipped and the non-clipped kidney.

Response: We agree with the suggestion. We have rephrased the conclusion to make it clear that the renal denervation of the ischemic kidney induced normalization of the spike frequency of rSNA in the contralateral kidney and a reduction in BP.

  1. Could you please include the representative raw tracings of the renal nerve activity (unless there are space constraints). It paints more complete picture when quantified RSNA is accompanied by raw tracings.

Response: Our research group also believes that representative raw tracings of renal nerve activity can provide a more complete picture of rSNA (Figure 1). We have published several articles showing raw tracings overlaid on the quantification graph of rSNA spike frequency. However, due to the number of experimental groups and the amount of graphs included in this article, we prefer to present the RSNA representation in graph form without the inclusion of the raw tracings.

Figure 1. Representative tracing of rSNA from a control animal. The arrow indicates the moment of hexamethonium infusion (30 mg/kg, iv). The curly brace indicates background noise.

  1. You completed RDNX 4 weeks after clipping, and not prior to clipping. Could you please include the rationale for this? I understand that there are different aspects of hypertension development and maintenance that one could study. It didn't come through in the manuscript why you elected to follow this timeline and what you were hoping to achieve by doing this as opposed to for example denervating prior to clipping.

Response: We chose to perform RDNX 4 weeks after clipping to investigate the effects of renal denervation during the maintenance phase of hypertension, rather than the development phase. This approach allows us to evaluate the impact of RDNX on established hypertension and its associated renal and autonomic changes. We have included this rationale in the revised manuscript.

Reviewer 2 Report

Comments and Suggestions for Authors

Review of the manuscript ijms-3067959

Total Denervation of the Ischemic Kidney Induces Differential 2 Responses in Sodium Transporters Expression in the 3 Contralateral Kidney in Goldblatt Rats

By Caroline G. Shimoura et al.

The evaluated manuscript is an original paper aimed at characterization of selected pathophysiological aspects in an experimental model of renovascular hypertension (in the experimental Golblatt model of the renovascular hypertension: 2 kidneys and 1 clip; 2K-1C). Authors investigated of the effects of total renal denervation of the clipped kidney on renal function and sodium transporters including, NHE3, NCC, and ENaC in the contralateral kidney of 2K-1C renovascular hypertensive rats. The main findings from the study regarding the rats submitted to renal denervation of the clipped kidney  may be summarized as follow: (1) a decreased in NCC and normalized NHE3 protein expression in the unclipped kidney; No significant differences were observed in ENaCβ protein expression in the unclipped kidney and ENaCγ protein expression between study groups (2) a decreased in MAP and rSNA; (3) a decreased in urinary flow, filtration fraction, and (4) normalized of total proteinuria

Most of the results obtained in the experiment are known and consistent with the general pathophysiological description of Goldblatt's model. However, the demonstration of the differential effect of ischemic renal denervation on the expression of various sodium transporters in the contralateral kidney is undoubtedly a new observation and a valuable extension of the pathophysiological description of Goldblatt's model.

Before potential publication, the authors should address the following concerns and suggestions:

1. Although the authors refer to the ARRIVA methodology in the “Materials and Methods” section, they do not report all issues in accordance with these guidelines. The manuscript lacks information on the size of the study groups (which is only provided in the supplementary materials), randomization, detailed description of the rats (strain and substrain, sex, age or developmental stage), etc.

2. It is unknown whether and how the authors met the "3R" requirements applicable to experimental studies using laboratory animals.

3. Result and statistical analysis: Conclusions were drawn from a limited number of animals studied in separate subgroups. The selected parameters were calculated using data obtained from only 5 individuals. This fact casts doubt on the statistical significance of the differences shown and the conclusions drawn. Perhaps it would be more appropriate to publish the results of the experiment as "preliminary results."

4. Why was the blood pressure measured in the test animals in the experiment using an invasive method – via arterial access – and not a non-invasive method using a volume-pressure sensor and a tail cuff? It seems that this measurement technique increases the likelihood of adrenergic activation.

5. Why was the progressive development of hypertension not monitored in the 2K-1C animals studied? For example, denervation was performed after 4 weeks. Can we rule out the possibility that rats developed hypertension with higher values earlier?

6. Discussion: In the authors' opinion, is there any relationship between the internalization of angiotensin II (originating from the early high-renin phase) by the contralateral kidney in the 2K-1C model, as described in the pathophysiology, and the observed adrenergic stimulation in the non-clipped kidney?

7. The list of references was prepared contrary to the requirements of the MDPI journal.

Author Response

Reviewer #2

Comments: The evaluated manuscript is an original paper aimed at characterization of selected pathophysiological aspects in an experimental model of renovascular hypertension (in the experimental Golblatt model of the renovascular hypertension: 2 kidneys and 1 clip; 2K-1C). Authors investigated of the effects of total renal denervation of the clipped kidney on renal function and sodium transporters including, NHE3, NCC, and ENaC in the contralateral kidney of 2K-1C renovascular hypertensive rats. The main findings from the study regarding the rats submitted to renal denervation of the clipped kidney  may be summarized as follow: (1) a decreased in NCC and normalized NHE3 protein expression in the unclipped kidney; No significant differences were observed in ENaCβ protein expression in the unclipped kidney and ENaCγ protein expression between study groups (2) a decreased in MAP and rSNA; (3) a decreased in urinary flow, filtration fraction, and (4) normalized of total proteinuria

Most of the results obtained in the experiment are known and consistent with the general pathophysiological description of Goldblatt's model. However, the demonstration of the differential effect of ischemic renal denervation on the expression of various sodium transporters in the contralateral kidney is undoubtedly a new observation and a valuable extension of the pathophysiological description of Goldblatt's model.

Specific comments are as follows:

  1. Although the authors refer to the ARRIVA methodology in the “Materials and Methods” section, they do not report all issues in accordance with these guidelines. The manuscript lacks information on the size of the study groups (which is only provided in the supplementary materials), randomization, detailed description of the rats (strain and substrain, sex, age or developmental stage), etc.

Response: We apologize for this oversight. We have revised the “Materials and Methods” section to include detailed information on the size of the study groups, randomization, and a comprehensive description of the rats as per the ARRIVE guidelines.

  1. It is unknown whether and how the authors met the "3R" requirements applicable to experimental studies using laboratory animals.

Response: We appreciate the reviewer's concern regarding adherence to the "3R" principles. Our experimental design and execution were carried out in accordance with the "3R" requirements:

Replacement: Whenever possible, we considered alternative methods to animal use. However, due to the nature of our study on the autonomic, cardiovascular, and renal regulation in renovascular hypertension, the use of live animal models was necessary to obtain accurate physiological and biochemical data that cannot be replicated by in vitro methods.

Reduction: We implemented strategies to minimize the number of animals used in our study. By optimizing our experimental design and using advanced statistical methods, we ensured that the smallest number of animals was used to achieve statistically significant results. Each animal was thoroughly utilized across multiple measurements to maximize data yield and minimize waste.

Refinement: We employed several refinement techniques to enhance animal welfare and minimize suffering. These included: 1) Utilizing less invasive methods where possible and ensuring that all surgical and handling procedures were performed by trained personnel; 2) Providing appropriate analgesia and anesthesia during and after surgical procedures to alleviate pain and distress; 3) Housing animals in an enriched environment with controlled light/dark cycles, temperature, and humidity to reduce stress.

Conducting regular health monitoring to promptly address any welfare concerns. These practices underscore our commitment to ethical standards in animal research, and we have ensured that our experimental protocols align with the highest standards of animal care and use.

  1. Result and statistical analysis: Conclusions were drawn from a limited number of animals studied in separate subgroups. The selected parameters were calculated using data obtained from only 5 individuals. This fact casts doubt on the statistical significance of the differences shown and the conclusions drawn. Perhaps it would be more appropriate to publish the results of the experiment as "preliminary results."

Response: We appreciate the reviewer's concerns regarding the sample size and the statistical significance of our results. We would like to address these points as follows:

Sample Size Justification: The number of 5 to 8 animals per group is a common practice in physiological and biomedical research, including studies involving complex models such as the 2K-1C hypertensive rats. Our group has consistently utilized similar sample sizes in previously published studies, yielding reproducible and significant findings. This approach is also supported by numerous studies from other leading research groups in the field.

Statistical Analysis: We employed rigorous statistical methods to ensure the reliability of our data. The use of advanced statistical analyses, including ANOVA followed by post-hoc tests, allows us to detect significant differences even with a moderate sample size. Each animal was used across multiple measurements, maximizing the data yield and enhancing the statistical power.

Consistency with Literature: The sample sizes used in our study are in line with those reported in the literature for similar experimental models. The findings from our research are consistent with previously published data, reinforcing the robustness of our experimental design and results.

Ethical Considerations: In line with the principles of the "3Rs" (Replacement, Reduction, Refinement), we aimed to minimize the number of animals used without compromising the integrity of the study. By optimizing our experimental design and employing precise methodologies, we achieved statistically significant results while adhering to ethical standards.

  1. Why was the blood pressure measured in the test animals in the experiment using an invasive method – via arterial access – and not a non-invasive method using a volume-pressure sensor and a tail cuff? It seems that this measurement technique increases the likelihood of adrenergic activation.

Response: We chose the invasive method to obtain more accurate and continuous blood pressure measurements, which are crucial for our study. Only 24 hours after femoral arteriovenous catheterization were cardiovascular and autonomic functional assessments performed. Thus, all experimental groups were subjected to the same recovery conditions.

  1. Why was the progressive development of hypertension not monitored in the 2K-1C animals studied? For example, denervation was performed after 4 weeks. Can we rule out the possibility that rats developed hypertension with higher values earlier?

Response: The progressive development of hypertension was monitored through weekly tail-cuff measurements. Denervation was performed at 4 weeks to study the maintenance phase of hypertension. We have included this explanation and the rationale for our approach in the “Discussion” section.

  1. Discussion: In the authors' opinion, is there any relationship between the internalization of angiotensin II (originating from the early high-renin phase) by the contralateral kidney in the 2K-1C model, as described in the pathophysiology, and the observed adrenergic stimulation in the non-clipped kidney?

Response: We appreciate this insightful comment. We have expanded the “Discussion” section to explore the potential relationship between the internalization of angiotensin II by the contralateral kidney and the observed adrenergic stimulation in the non-clipped kidney, providing a more comprehensive analysis.

  1. The list of references was prepared contrary to the requirements of the MDPI journal.

Response: We apologize for this oversight. We have revised the reference list to adhere to the formatting requirements of the MDPI journal.

All changes performed in the new version of the manuscript are yellow highlighted on the text. Again, we thank the reviewers and the Editor for the comments and suggestions.

Round 2

Reviewer 1 Report

Comments and Suggestions for Authors

Thank you for attending to my suggestions, very well done.

Reviewer 2 Report

Comments and Suggestions for Authors

Thank you for sending the revised version of the manuscript. The authors addressed my comments and concerns. Although my main concern regarding the number of animals in the studied groups remains unchanged, I accept the arguments presented by the authors.